# Mechanisms Involved in the Neurotoxicity and Abuse Liability of Nitrous Oxide: A Narrative Review

**DOI:** 10.3390/ijms232314747

**Published:** 2022-11-25

**Authors:** Tibor M. Brunt, Wim van den Brink, Jan van Amsterdam

**Affiliations:** Department of Psychiatry, Amsterdam UMC, Location Academic Medical Center, University of Amsterdam, P.O. Box 22660, 1100 DD Amsterdam, The Netherlands

**Keywords:** nitrous oxide, N_2_O, dependence, neurotoxicity, cobalamin, dopamine, abuse

## Abstract

The recreational use of nitrous oxide (N_2_O) has increased over the years. At the same time, more N_2_O intoxications are presented to hospitals. The incidental use of N_2_O is relatively harmless, but heavy, frequent and chronic use comes with considerable health risks. Most importantly, N_2_O can inactivate the co-factor cobalamin, which, in turn, leads to paresthesia’s, partial paralysis and generalized demyelinating polyneuropathy. In some patients, these disorders are irreversible. Several metabolic cascades have been identified by which N_2_O can cause harmful effects. Because these effects mostly occur after prolonged use, it raises the question of whether N_2_O has addictive properties, explaining its prolonged and frequent use at high dose. Several lines of evidence for N_2_O’s dependence liability can be found in the literature, but the underlying mechanism of action remains controversial. N_2_O interacts with the opioid system, but N_2_O also acts as an N-methyl-D-aspartate (NMDA) receptor antagonist, by which it can cause dopamine disinhibition. In this narrative review, we provide a detailed description of animal and human evidence for N_2_O-induced abuse/dependence and for N_2_O-induced neurotoxicity.

## 1. Introduction

Over the past decade, the prevalence of recreational nitrous oxide (N_2_O) use has increased in the Western world [1]. For instance, the 2019 Global Drug Survey (GDS), an online drug survey among a self-selected sample of drug users from over 30 countries, showed that 91% of all participants (*n* = 123,814) had used N_2_O at least once, suggesting that N_2_O is the 10th most popular drug, excluding alcohol and tobacco, in the Western world [2]. According to the 2019/2020 ‘Household Survey’, the highest rate of last year’s N_2_O use in adolescents (16–24 years) in the United Kingdom was 8.7% in England and Wales, which implies that N_2_O was second only to cannabis in use among those aged 16–24 in England and Wales [3]. N_2_O use also seems to have increased in the United States [4]. Similarly, among French students, N_2_O use comes in second place after cannabis; last year, the rates were 14% and 35%, respectively [5]. Interestingly, N_2_O use in the Netherlands is especially high among young non-Western immigrants, such as second generation immigrants from Morocco or Turkey, aged 12–16 years, with rates of ever use and last-month use of 12.8% and 3.9%, respectively (their Dutch peers: 8.9% and 2.0%, respectively) [6]. However, the public health consequence of this widespread N_2_O use is low, because N_2_O is a relatively safe drug when used only occasionally and in low doses [7]; typical recreational users consume less than 10 bullets (‘whippets’; each containing 10 mL of pressurized N_2_O) per session.

However, recently, the number of young excessive users has risen—for instance, among young non-Western immigrants [6]. Likewise, the Dutch Poisons Information Center reported a steep increase in N_2_O intoxications from 0.12% in 2010 to 11% in 2020, with an average monthly rate of 3.8% of all reported intoxications [8], 79% of the patients indicating heavy and frequent use in 2019 and 2020 and 42% using N_2_O from large cylinders. These alarming increases in N_2_O abuse or intoxications have also been reported in other countries during the same period, such as Australia [9,10], the United States [4], China [11] and France [12]. N_2_O is used in largely the same way across all these countries, with whippets (or bullets) being the most prevalent. However, the use from larger canisters has also been seen more recently, such as in the United States [13]. Frequent and heavy use (up to 700 whippets per day) has been reported in Australia [14]. This is of serious concern because repeated exposure to high doses of N_2_O for a prolonged time is known to induce neurological damage, such as (irreversible) neuropathy and paralysis due to N_2_O-induced cobalamin deficiency [15,16,17,18,19]. The increasing trend of recreational users with N_2_O-induced neurological damage at emergency departments confirms the urgency of this development [9,20,21].

In some recreational N_2_O users, N_2_O abuse and/or dependence seems to develop with craving, a loss of control and continued use despite social and/or physical damage. Obviously, psychosocial factors are important factors in the development of excessive N_2_O use, abuse and dependence [6]. For instance, among young non-Western Muslim immigrants, marginalization, boredom, unemployment, deteriorated social interactions, social isolation, macho behaviour, shame and distrust of the Dutch medical system appeared to be important drivers of N_2_O abuse [6]. In addition, the recent introduction and availability of larger tanks or ‘smart whip’ cylinders, containing 0.6–10 kg of N_2_O, have certainly facilitated higher repeated N_2_O dosing and N_2_O abuse in the Netherlands [8] and France [22]. Others, such as those in France or China, claim that N_2_O abuse has increased during the COVID-19 pandemic due to boredom as a result of the lock-downs [12,18,23], although evidence for this claim is rather weak.

Whereas there have been many case reports, some case-series and a number of reviews about N_2_O intoxication and its supposed mechanisms, relatively few studies have been conducted on the dependence potential of N_2_O. Additionally, there seems to be some controversy about the mechanisms by which N_2_O might induce abuse (binge use) or dependence, as was recently illustrated by an article of Kamboj et al. [24], showing that N_2_O rewarding effects are mainly mediated through the blockade of the N-methyl-D-aspartate (NMDA) receptors by N_2_O. This was followed by a comment in the same journal by Gillman [25], who has done pioneering work on the neuropsychological mechanisms of action of N_2_O in the past, stipulating that N_2_O’s actions were more likely due to its ability to interact with the opioid system. Therefore, in this review, we will give an overview of the animal and human evidence on the mechanisms of action involved in N_2_O-induced neurotoxicity, which may arise from N_2_O-induced abuse/dependence (frequent and prolonged use), two issues that were never combined in previous reviews.

## 2. Neurotoxicity of N_2_O

### 2.1. Acute Neurotoxicity

In a meta-analysis on hospitalized cases presented after N_2_O exposure, the most frequent clinical symptoms were paresthesia (80%), unsteady gait (58%) and limb weakness (43%) [17]. Similar clinical symptoms were reported by other (clinical) studies, as well as paraplegia, numbness and vestibular problems [8,26,27]. In a global drug user survey, alongside these clinical symptoms, mental symptoms were reported, such as hallucinations and confusion [28]. When used sporadically, about 3% of the users reported paresthesia [26,27,29].

### 2.2. Chronic Neurotoxicity

The chronic use of N_2_O has been associated with serious consequences, such as peripheral neuropathy, myelopathy and demyelizing diseases, collectively referred to as generalized demyelinating polyneuropathy (GDP) [17]. This is expressed in clinical symptoms such as muscle weakness, vestibular disturbances and paralysis [30]. Recent MRI studies showed the progressive degeneration of the spinal cord in N_2_O users [31]. A correlation was found between the extent of N_2_O use (in whippets or balloons) and the degree of myelopathy and GDP [32], and most chronic users (mean: 300 balloons/day for 6 months) displayed signs of neuropathy. Cobalamin deficiency in patients with GDP has been a common finding in a number of studies [16,17,27], and cobalamin (vitamin B12) supplementation induces substantial neurological improvement or even recovery in most patients [27]. Nonetheless, some of these patients will only partly recover, with persistent neuropathies, such as paresthesia’s, limb weakness and/or partial paralysis, and are therefore in continuous need of medical devices [14,33]. Furthermore, chronic N_2_O use has been associated with psychiatric symptoms, such as anxiety, depression, neurocognitive deficits and delirium [15]. However, these psychiatric symptoms did not seem to result from cobalamin deficiency [34].

## 3. The Molecular Mechanisms behind N_2_O Neurotoxicity

Although cobalamin has been found to be decreased in chronic N_2_O users with neurological damage, this is not always the case, making it unlikely that vitamin B12 deficiency is the only cause of neurological damage. In fact, many studies in chronic N_2_O users did not find a correlation between cobalamin levels and neurological damage [14,26,31]. In fact, elevated serum levels of homocysteine and methylmalonate (methylmalonic acid) were better biomarkers for the neurological damage after prolonged N_2_O exposure [14,16,26]. This raises the issue of which metabolomic mechanisms are exactly involved in N_2_O-induced toxicity.

At the core of the neurological damage associated with chronic N_2_O use lies a disturbance of cobalamin metabolism [35]. Cobalamin functions as an essential co-factor in the regeneration of methionine and the formation of tetrahydrofolate, which is involved in biosynthetic pathways of nucleic acid and amino acid metabolism [36,37]. N_2_O induces irreversible oxidation of the cobalt-ion in cobalamin, whereby it no longer functions, as the co-factor methylcobalamin, in the enzymatic formation of methionine and tetrahydrofolate. DNA/RNA/protein methylation by methionine is an essential step in the production of phospholipids of the myelin sheath [38]. Disrupted DNA/RNA methylation by decreased levels of methionine, through the oxidation of cobalamin by N_2_O, has been implicated in many neurodegenerative disorders [39,40]. A grieve medical condition that has been ascribed to N_2_O exposure is subacute combined degeneration, which is characterized by the degeneration of the spinal cord columns due to demyelination, presented through paresthesia, weakness, ataxia, gait disturbance and, if untreated, paraplegia [33,41]. This is caused by the accumulation in the myelin sheath of other substrates of cobalamin, such as methylmalonate [42]. Animal studies found that methylmalonate accumulation (methylmalonate aciduria) is neurotoxic [43,44]. Methylmalonic aciduria in rat striatal neurons resulted in the inhibition of respiratory chain complex II, the tricarboxylic acid cycle, toxic organic acids and synergistic secondary excitotoxic mechanisms [45,46]. Methylmalonate is a precursor in the biochemical conversion of methylmalonyl-CoA to succinyl-CoA by methylmalonyl-CoA mutase, and this conversion is blocked by the oxidation of cobalamin by nitrous oxide. A schematic overview of some of the main toxic mechanisms of N_2_O is depicted in Figure 1.

Whereas these factors are proposed to be at the center of N_2_O neurotoxicity, other contributing factors have also been proposed. Homocysteine, which accumulates under chronic N_2_O exposure, can be neurotoxic, causing the overstimulation of N-methyl-D-aspartate (NMDA) receptors, leading to an increase in cytoplasmic calcium ions and the accumulation of reactive oxygen species (oxidative stress), causing apoptosis [47,48]. N_2_O itself is a noncompetitive antagonist of the NMDA receptor, which would result in a neuroprotective effect, but this may only be on the short term [49,50]. A prolonged blockade by N_2_O might result in neuronal vacuolation [11]. Disrupted methylation, by the oxidation of cobalamin, has also been linked to the deleterious functioning of the immune system by a reduced proliferation of lymphocytes on one hand and cobalamin’s direct effects on cytokines and growth factors on the other [36,51].

Taken together, (frequent) exposure to N_2_O leads to the inactivation of cobalamin, the blockade of NMDA receptors and a cascade of metabolic effects that all contribute to neurotoxicity, which is responsible for the adverse events observed in the clinical practice, such as spinal cord degeneration.

## 4. From Incidental Use to N_2_O Abuse (Binging)

When looking at N_2_O’s harmful effects, it seems that there is a great disparity between incidental N_2_O exposure and prolonged N_2_O exposure in its propensity to generate toxic effects [7,8]. This raises the issue of whether N_2_O can induce a pattern of frequent use, binging, abuse or even dependence. Several studies have tried to resolve this issue throughout the years; efforts have been undertaken to explain the abuse/dependence potential of N_2_O. We will discuss this below.

## 5. Dependence Liability

### 5.1. Human Data

The abuse and dependence liability of N_2_O is a currently underexposed and poorly investigated topic. N_2_O is often used repetitively in one session, mainly due to its short half-life of approximately 5 min [52]. However, highly frequent sessions of N_2_O use with a longer duration (several hours to all day with 150–700 10 mL bullets used daily) for several days have also been described, suggesting that N_2_O may have a dependence potential [11,14,29]. Interestingly, anecdotal evidence (Sebastiaan Verboeket, Jellinek Addiction Clinic, The Netherlands, personal communication) indicates that N_2_O abusers are more or less binging N_2_O, i.e., they heavily use N_2_O for 3–5 days (mostly using 2 kg tanks), refrain from using N_2_O for 3–6 weeks and restart heavy N_2_O use for some days again. Moreover, some users maintain such high dosing despite N_2_O-related physical harm, which is another hallmark of dependence. Unlike with other substances of abuse, N_2_O abuse does not cause direct physical withdrawal symptoms upon the acute cessation of N_2_O use [53]. For this reason, N_2_O and other short-lived inhalants were originally classified as a separate group of abused substances without a dependence potential [54], mainly because N_2_O does not cause withdrawal symptoms, which is typical for substances of dependence. In the fifth edition of the Diagnostic and Statistical Manual of Mental Disorders (DSM-5), N_2_O use disorder was categorized under “other substance use disorders” [55], indicating that it is not a substance of dependence.

However, N_2_O is regarded as a substance of dependence by some [56,57], despite the fact that the evidence for this is not unambiguous. For instance, human volunteers did not favor the inhalation of N_2_O (in concentrations ranging from 20–80%) over oxygen [58,59], indicating a lack of reinforcement effects (craving) of N_2_O, and both brief and extended exposures to N_2_O yielded the same results. In a more recent study based on the information of 59 subjects who had used N_2_O in larger quantities and for longer than intended, Fidalgo et al. [60] identified, in 2019, an ‘N_2_O use disorder’ according DSM-5 criteria and suggested that N_2_O has a low degree of dependence potential. Their data suggest just a mild substance use disorder (SUD), as only two to three *DSM*-5 criteria were met. Whereas N_2_O lacks reinforcing effects in humans, tolerance for its analgesic effects was found [61].

Finally, it appeared that N_2_O, at subanaesthetic concentrations, acts as an opioid receptor agonist [25]. Interestingly, naltrexone, an opioid receptor antagonist which is used for treating opioid and alcohol dependence, was reported to be an effective treatment in a case of N_2_O abuse [62]. Interestingly, naltrexone also was proven effective in a case of ketamine dependence, a substance with similar anaesthetic applications as N_2_O [63]. However, being a partial opioid agonist (see below), it has, by definition, a lower dependence liability than full opioid agonists, such as morphine or heroin. Recreational N_2_O use is routinely practiced at subanaesthetic doses, i.e., users remain fully conscious, indicating that the possible dependence potential of N_2_O might be linked to the opioid receptor agonism.

Summarizing the human evidence, N_2_O does not seem to fulfil the traditional criteria for a substance of dependence, as it is not associated with withdrawal and lacks reinforcing effects in clinical human studies. However, tolerance for its effects occurs, and the abuse potential for N_2_O was shown, with 2 to 3 criteria of the definition of a substance of dependence being met, indicative of a mild N_2_O use disorder, which would be a more appropriate term.

### 5.2. Animal Studies

So far, there are limited animal studies (e.g., self-administration, drug discrimination) about compulsive N_2_O use or N_2_O abuse/dependence. Typical for addictive substances is that they show self-administration, induce tolerance and show withdrawal upon the acute cessation of heavy use.

To begin with, self-administration studies in animals may give evidence for reinforcing properties of a substance. However, conflicting results have been obtained in such studies following the administration of N_2_O. In one conditioned place preference study in rats, N_2_O failed to induce reinforcement [64]. In another study, the intracranial self-stimulation of N_2_O in mice showed a mild reinforcing effect [65]. In squirrel monkeys, N_2_O could be self-administered by pressing a key, which showed a progressive administration ratio in comparison to controls [66], indicating that nitrous oxide can function as a reinforcer.

Tolerance for the effects of a drug is another criterium of a substance of dependence. In animal and human studies, tolerance for the analgesic effects of N_2_O was proven [61,67], which is some evidence for N_2_O being a substance of dependence. Furthermore, tolerance was also shown in a study on N_2_O-induced locomotion and visual-evoked potentials (VEP) in rats [68]. Withdrawal is another criterion needed for physical dependence. As such, the acute cessation of chronic exposure to N_2_O is expected to elicit in signs of withdrawal such as excitation, psychomotor stimulation, convulsions and hypertension. Indeed, during N_2_O withdrawal, mice convulsed when gently lifted by the tip of the tail [69,70]. For instance, the cessation of the exposure of mice to nitrous oxide (at 50, 65 and 80% for 34 or 68 h) resulted in characteristic dose- and exposure time-dependent convulsions very similar to those seen in alcohol-dependent mice upon withdrawal. Convulsions were maximal within 2–3 min after the cessation of N_2_O use and declined over 6 h [71]. Other studies found stress in rats during N_2_O withdrawal, which was linked to decreases in beta-endorphin [72], and N_2_O exposure blocked morphine-induced conditioned place preference [73].

Taken together, animal studies show contrasting evidence for a dependence potential of N_2_O. It has some degree of tolerance and withdrawal but only a low reinforcing activity. Given these uncertainties, it remains dubious as to whether N_2_O is a typical addictive substance.

## 6. Molecular Mechanisms of N_2_O Abuse and Dependence

As proposed and further elaborated by the group of Gilman, N_2_O exerts its analgesic actions via interaction with the opioid system [25,74,75]. N_2_O activates opioid neurons in the brainstem, relieving pain throughout the central nervous system [76]. Endogenous opioid activation in the brainstem inhibits gamma-aminobutyric acid (GABA)-releasing neurons, in turn activating descending noradrenergic pathways that inhibit pain [52,56,57,75,77,78,79,80]. It was found that the antinociceptive effects of N_2_O are likewise mediated through the adrenergic α1 and α2-receptors in the spinal cord [76].

The mechanisms by which N_2_O interacts with the opioid system have been under investigation for several decades. It was thought that N_2_O exerts its antinociceptive—and possibly also addictive—effects mainly as a partial agonist of the opioid receptors [56]. In studies that followed, the κ-opioid receptor was identified as the main target for N_2_O’s antinociceptive effects, as selective κ-receptor antagonists and spinal pretreatment with antiserum directed against the endogenous κ-receptor ligand dynorphin blocked N_2_O’s antinociceptive effects [52]. An important finding was that of the cross-tolerance to N_2_O of morphine-tolerant rodents [81]. This cross-tolerance was unidirectional, because morphine (and other opioids) still produced antinociception in N_2_O-tolerant rodents [82], meaning that the responsiveness of the opioid receptors was not altered [52]. In this regard, N_2_O seemed to act in more ways than merely an opioid receptor agonist, considering that N_2_O is able to release endogenous opioids directly from the periaqueductal area in the midbrain [75,82,83,84]. Regarding N_2_O’s possible potential to induce abuse or dependence, the mechanism of the release of endogenous opioids makes the most sense, as dependence is mainly mediated through the μ-opioid receptor and not the κ-opioid receptor [85].

Another mechanism of the possible addictive properties of N_2_O is its antagonism at the NMDA-receptor [49]. Like ketamine, another NMDA-receptor antagonist, it is used as both an anaesthetic and ketamine is a substance of abuse with a proven risk of dependence [86]. The mechanism that is proposed for the rewarding properties of these anaesthetics is that the blockade of NMDA-receptors uplifts the inhibition on dopamine neurons by GABAergic neurons, especially in the ventral tegmental area and the nucleus accumbens, creating dopamine burst firing [87,88], an effect also demonstrated in humans through brain imaging [89]. In 1983, Hynes and Berkowitz showed that haloperidol inhibited N_2_O-induced locomotor activity in mice and showed direct involvement of the dopaminergic system, while ten years later, the depletion of catecholamine synthesis was also reported to block N_2_O-induced locomotor activity [52,77]. Subsequent studies showed region-dependent effects of N_2_O on dopamine and/or noradrenaline levels or turnover in the brain following the exposure of rats to N_2_O. Thus, considering that ketamine produces strong psychotomimetic effects, it was suggested that the euphoric effects induced by N_2_O are (at least partly) due to the similar inhibition of NMDA receptor-mediated neural substrates [90]. There is some debate on the issue of at which levels of N_2_O exposure these effects occur [49], but a recent study indicates that N_2_O shows ketamine-like excitatory effects at subanesthetic but therapeutically relevant concentrations [91]. Figure 2 shows a schematic overview of the putative mechanisms of action of N_2_O on dependence and abuse.

## 7. Conclusions

N_2_O affects various biomolecular pathways that are possibly relevant to its abuse potential and thereby contribute to its neurotoxicity. However, both animal and human studies investigating whether or not N_2_O is actually able to induce dependence, as was also defined by the DSM-5, provided inconclusive data. Based on the available literature, a mild N_2_O use disorder seems to be the most appropriate term. Mechanistically, there are some modes of action that are described by which N_2_O could induce this abuse/mild dependence potential. N_2_O does seem to release opioids from the periaqueductal grey area that interact with GABAergic neurons in the midbrain, disinhibiting dopamine release. This seems like a plausible mechanism by which N_2_O could cause reward and craving. However, NMDA-antagonism, as another explanation, cannot be ruled out, as this also disinhibits dopamine release in the ventral tegmental area and the nucleus accumbens, similar to how another anaesthetic and recognized addictive substance, ketamine, works. Both mechanisms are not mutually exclusive and most likely reinforce each other. Previously, it was thought that NMDA-antagonism only occurred at anaesthetic levels, but recent evidence showed that this mechanism also occurs at subanaesthetic levels in recreational and frequent N_2_O users.

Frequent N_2_O abuse gradually inactivates cobalamin, with the degeneration of the spinal cord being a possible consequence, mainly through the disrupted DNA/RNA/protein methylation needed for the production of phospholipids of the myelin sheath. Cobalamin deficiency-induced homocysteine accumulation adds to the N_2_O-induced neurotoxicity by NMDA receptor overstimulation, and homocysteine and methylmalonic acid are the most consistent clinical biomarkers of chronic N_2_O abuse and intoxication. When detecting early signs of N_2_O toxicity (such as paraesthesia and numbness), this calls for systematic screening for those biomarkers when considering N_2_O-related toxicity or abuse/dependence, perhaps in conjuncture with a spinal cord MRI. Besides neurotoxicity, other N_2_O-related toxicity has also been documented, such as adverse reproduction effects in females after N_2_O exposure [92]. For N_2_O-related neurotoxicity, most symptoms can be reversed by vitamin B12 suppletion.

For suspected N_2_O-related substance use disorder, current state-of-the-art dependence therapy can be offered, possibly with special attention to non-Western immigrants [6]. Furthermore, opioid antagonism can be considered, as naltrexone therapy proved effective in a case of N_2_O dependence. In accordance, patients displaying early symptoms of N_2_O toxicity should be educated by physicians and addiction professionals about the potentially dangerous consequences of prolonged heavy N_2_O use to prevent further, irreversible harm and emerging N_2_O use disorders.

## Figures and Tables

**Figure 1 ijms-23-14747-f001:**
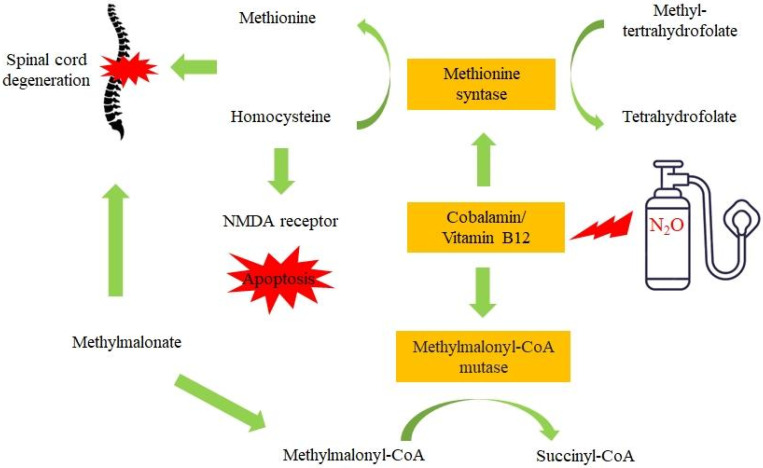
The main mechanisms involved in N_2_O-induced neurotoxicity. Central toxicity arises from the deactivation of the cobalt-ion at the cobalamin (vitamin B12) molecule, which leads to a cascade of neurotoxic effects.

**Figure 2 ijms-23-14747-f002:**
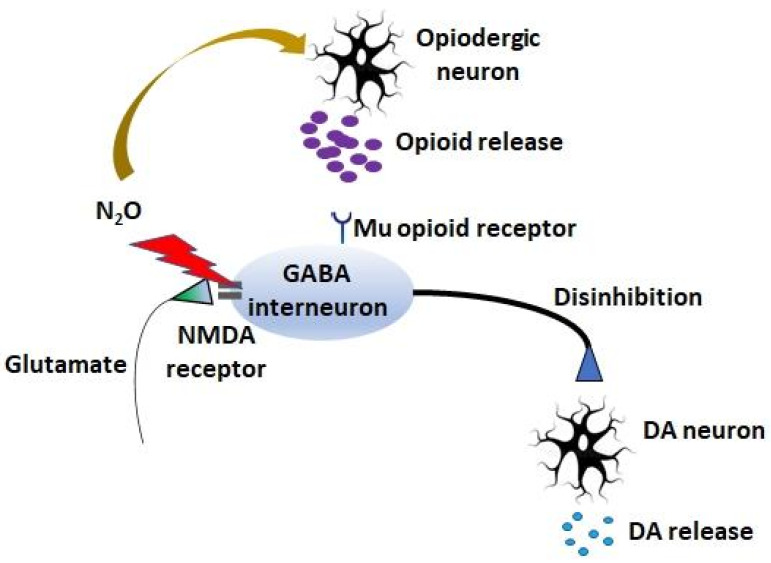
The known mechanisms of action of N_2_O that are involved in abuse and dependence. N_2_O induces opioid release in the periaqueductal grey area, on the one hand, and it also acts as an N-methyl-D-aspartate (NMDA) receptor antagonist, on the other hand. Via both routes, N_2_O is able to inhibit gamma-aminobutyric acid (GABA) interneurons, disinhibiting dopamine (DA) release, which can cause symptoms of abuse and dependence.

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
