# Peer review of "Mechanisms Involved in the Neurotoxicity and Abuse Liability of Nitrous Oxide: A Narrative Review"

_ijms, 2022, doi:10.3390/ijms232314747_

Round 1

Reviewer 1 Report

The review is written very clearly and in good English. However, this is a summary of already well-known facts. The author did not come up with anything new, with any new interpretation. But it probably wasn't even the goal.

1) On the formal side, there were two minor errors:

lines 141 to 143: "Elevated serum levels of homocysteine and methylmalonic acid are the most consistent clinical biomarkers associated with chronic N2O exposure [15,26,27,31]" belongs to the previous paragraph.

line 167: Title "Dependence liability" should have different formatting, and this is chapter 5, then renumber the next two.

2) And then I have a comment regarding the interpretation of data from one preclinical study:

lines 223 to 225: You claim that the study (Wood et al., 1977) shows that N2O is a reinforcer based on quitting pressing the lever in the absence of the drug. It is not like that. The positive reinforcing effect of the substance is demonstrated in this model based on the progressive ratio. The extinction phase (the animal is in the box but without the drug) is performed to prove relapse (after reinstatement).

Author Response

The review is written very clearly and in good English. However, this is a summary of already well-known facts. The author did not come up with anything new, with any new interpretation. But it probably wasn't even the goal.

-answer authors-

Thank you for the kind compliments and our goal was mainly to give a full oversight of nitrous oxide in terms of dependence liability and subsequent neurotoxicity arising from frequent use. Most reviews only shed light on one aspect, but we’ve taken them altogether.

1) On the formal side, there were two minor errors:

lines 141 to 143: "Elevated serum levels of homocysteine and methylmalonic acid are the most consistent clinical biomarkers associated with chronic N2O exposure [15,26,27,31]" belongs to the previous paragraph.

-answer authors-

We agree with the reviewer and have deleted it from this paragraph. There was already a sentence about homocysteine and methylmalonic acid in the first paragraph.

line 167: Title "Dependence liability" should have different formatting, and this is chapter 5, then renumber the next two.

-answer authors-

Apparently, this has been deleted while formatting for the journal, we have correctly formatted and numbered this in the revision.

2) And then I have a comment regarding the interpretation of data from one preclinical study:

lines 223 to 225: You claim that the study (Wood et al., 1977) shows that N2O is a reinforcer based on quitting pressing the lever in the absence of the drug. It is not like that. The positive reinforcing effect of the substance is demonstrated in this model based on the progressive ratio. The extinction phase (the animal is in the box but without the drug) is performed to prove relapse (after reinstatement).

-answer authors-

We agree with the reviewer and have modified this sentence (In squirrel monkeys N2O could be self-administered by pressing a key, which showed a progressive ratio in comparison to the controls).

Reviewer 2 Report

The authors perform an extensive review on the recreational use of nitrous oxide.

In an extensive and detailed introduction, they explain the consumption habits of nitrous oxide, focusing mainly on European studies.

Further on, nitrous oxide toxicity, clinical manifestations, and the molecular mechanisms underlying said toxicity are discussed.

The final part of the work tells us about the addiction to nitrous oxide and the mechanisms underlying said addiction.

This is a very complete review, and despite the fact that the contributions in some sections such as those referring to mechanisms of toxicity or consumption habits have already been described in detail in previous works (10.3390/brainsci4010073 and works by the authors of this manuscript), the contributions in the other sections make this review novel.

I suggest that the work be published in your magazine, and I only have two minor points.

a) The authors could mention fertility problems in women exposed to nitrous oxide (in fact they have a paper on the matter).

b) Data could be provided (if available) on consumption habits in other countries.

c) Despite the title, little text is devoted to explaining the mechanisms of toxicity or addiction

Author Response

The authors perform an extensive review on the recreational use of nitrous oxide.

In an extensive and detailed introduction, they explain the consumption habits of nitrous oxide, focusing mainly on European studies.

Further on, nitrous oxide toxicity, clinical manifestations, and the molecular mechanisms underlying said toxicity are discussed.

The final part of the work tells us about the addiction to nitrous oxide and the mechanisms underlying said addiction.

This is a very complete review, and despite the fact that the contributions in some sections such as those referring to mechanisms of toxicity or consumption habits have already been described in detail in previous works (10.3390/brainsci4010073 and works by the authors of this manuscript), the contributions in the other sections make this review novel.

 I suggest that the work be published in your magazine, and I only have two minor points.

-answer authors-

We want to thank the reviewer for these kind words and compliments. We are aware that certain aspects were covered extensively before, but have tried giving it some novelty. However, we did include the review you suggested as well.

  1. The authors could mention fertility problems in women exposed to nitrous oxide (in fact they have a paper on the matter).

-answer authors-

Indeed we have covered this in another study, we have mentioned this now in the discussion, as the effects on fertility are somewhat beyond the scope of our present review.

  1. Data could be provided (if available) on consumption habits in other countries.

-answer authors-

Agreed and we have included some studies and observations from other countries to make the introduction more comprehensive. But there are very few in-depths studies on habits or motives.

  1. Despite the title, little text is devoted to explaining the mechanisms of toxicity or addiction

-answer authors-

We are not entirely sure what is meant by this comment, we have devoted a large part of the review to the mechanisms of toxicity and addiction.